# Guided Bone Regeneration Using Carbonated Apatite Granules and L-Lactic Acid/ε-Caprolactone Membranes: A Case Series and Histological Evaluation

**DOI:** 10.3390/dj13020085

**Published:** 2025-02-17

**Authors:** Yoichi Taniguchi, Tatsuro Koyanagi, Yutaro Kitanaka, Azusa Yamada, Akira Aoki, Takanori Iwata

**Affiliations:** 1Department of Periodontology, Graduate School of Medical and Dental Sciences, Institute of Science Tokyo, Tokyo 113-8549, Japan; 2Taniguchi Dental Clinic, Sapporo 003-0023, Japan; 3Kudan Dental Clinic, Tokyo 113-8549, Japan; 4Oral Diagnosis and General Dentistry, Institute of Science Tokyo Hospital, Tokyo 113-8549, Japan

**Keywords:** guided bone regeneration, poly (L-lactic acid/ε-caprolactone) membrane, carbonated apatite, ridge augmentation, histological study, case series

## Abstract

**Background/Objectives**: The newly developed poly L-lactic acid/ε-caprolactone (P(LA/LC)) membrane has recently been proposed as a guided bone regeneration (GBR) procedure in implant treatment. This case series details the clinical, radiographic, and histological results of ridge augmentation using bone graft comprising a P(LA/LC) membrane and carbonated apatite for implant treatment. **Methods**: Ten patients (15 sites) requiring bone augmentation and implant placement were retrospectively assessed. Simultaneous implant placement (Si) was performed at five sites in three patients. Si immediately following tooth extraction (SiIP) was performed at four sites in four patients. The staged approach (St) was performed at six sites in three patients. In the St-treated cases, hard tissue samples were taken from the implant placement site under patient consent. **Results**: The mean regenerated bone width for each treatment method was Si, 6.34 ± 2.64 mm (excluding implant diameter, 2.60 ± 2.42); SiIP, 7.55 ± 1.17 mm (excluding implant diameter, 3.90 ± 0.78) and St, 5.57 ± 1.08 mm. The mean regenerated bone width for all the cases was 6.36 ± 1.83 mm (excluding implant diameter, 4.14 ± 1.99). Significant differences were observed between the pre- and post-operative bone width in all the cases and the SiIP group (*p* < 0.001). All cases were followed up for more than 2 years after attaching the superstructure. No inflammation, shrinkage, or other problems were observed in the hard and soft tissues surrounding the implant. In the histological evaluation, there was no soft tissue ingrowth into the augmented bone, and new bone formation was observed. **Conclusions**: The use of P(LA/LC) membranes and carbonated apatite as GBR materials in implant treatment resulted in stable and favorable bone augmentation.

## 1. Introduction

In the case of implant treatment, membrane techniques are applied as guided bone regeneration (GBR) techniques with sufficient treatment outcomes reported in clinical research [1,2]. GBR technique uses resorbable membranes reducing the number of surgeries. However, the use of a resorbable membrane, especially the use of early resorbable membranes such as polylactic acid membranes, and the use of membranes that are not tightly anchored to the bone during the GBR procedure has limited effect on alveolar bone augmentation, owing to the lack of space maintenance ability and unpredictable resorption which can cause the collapse of bone augmentation space [3]. On the other hand, non-resorbable membranes made with dense or expanded polytetrafluoroethylene (PTFE) and bone graft substitutes have been used for the regeneration of large bone defects [4]. In the case of enormous bone defects, a titanium frame-reinforced PTFE membrane was used to stabilize the bone graft material and prevent its displacement [5,6]. However, complication from early exposure was a disadvantage of using the PTFE membrane, and additional surgery was required for the removal of the PTFE membrane [7]. Thus, resorbable membranes have been used in clinical practice in recent years. Urban et al. reported the regeneration of an enormous bone defect using elastic thin collagen; this technique was termed the sausage technique [8]. Modification of the sausage technique resulted in the development of the tent screw-pole technique, wherein tightly packed bone graft substitutes are reinforced by elastic resorbable collagen membranes and bone tacks [9]. With the introduction of this technique, vertical and large horizontal ridge augmentation could be performed without a titanium-reinforced PTFE membrane or tent pole bone screw [3].

Recently, a new membrane material, poly (L-lactic acid/ε-caprolactone) (P(LA/LC), has been introduced. As it is a fully synthetic material, concerns regarding the use of biomaterials are also reduced [10]. The resorption period has been reported to be optimal for bone augmentation by using P(LA/LC) as the material. Moreover, this membrane has characteristics of an extensibility resorbable membrane and is slightly thicker and stronger than non-crosslink collagen membranes. Despite these optimal characteristics for GBR, clinical reports are limited. Although the clinical results of GBR using this membrane in terms of bone augmentation using a staged approach have been reported in several manuscripts, the clinical results for a variety of techniques, such as GBR for immediate extraction implant placement, have not been reported. On the other hand, carbonated apatite (CA) is a recently developed product for bone regenerative and augmentation procedure. CA is a chemical substance of the major human bone in organic component. In contrast to CA is completely remodeled to the bone tissue with a longer period than that of β-tricalcium phosphate (β-TCP). This delayed resorption property is considered for suitable bone regeneration. Moreover, compared with bovine bone mineral (BBM), CA has characteristics of early-stage new bone formation and a higher rate of remodeling to bone tissue in vivo study [11,12,13,14,15]. These characteristics lead to better clinical outcomes following sinus floor elevation compared to other synthetic bone graft materials [13]. Moreover, although CA and BBM have different basic micro structures—CA has a full structure, whereas BBM has a porous structure—they do not produce different clinical results in terms of the regenerated bone volume [16]. In addition, while Taniguchi et al. did not require mixing of CA with autogenous bone, Shido et al. did [16,17]. These reports do not suggest the need for autogenous bone augmentation when CA is used. Therefore, the purpose of this study was to clinically, radiologically, and histologically evaluate cases of GBR using this membrane and CA alone under various clinical conditions to evaluate their efficacy in clinical applications.

## 2. Materials and Methods

### 2.1. Case Series

The clinical data used in this case report were de-identified, collected, and analyzed. The cases included 15 sites in 10 patients (6 males and 4 females, mean age 55.6 ± 14.3 years) who underwent extensive bone augmentation using GBR as the amount of remaining bone in each implant site was insufficient for implant placement (Table 1). All participants showed improvement in bone quantity using the GBR technique. The exclusion criteria for surgery were as follows: patients with psychiatric disorders, smokers, pregnant women, and patients with systemic diseases that could affect surgical outcomes (diabetes, cardiovascular disease, cerebrovascular disease, osteoporosis, and taking bisphosphonate drugs). Participants were thoroughly informed of the risks associated with the procedures and provided written informed consent. Clinical data were obtained from medical records 12 months after bone regenerative therapy.

The study was conducted in accordance with the Declaration of Helsinki and Ethics Committee approval for the retrospective study. Approval for the clinical study was obtained from the Japanese Society of Oral Implantology (protocol code, 11000694; date of approval, 27 December 2023). Approval for the histological evaluation of bone augmentation was obtained from the Review Board of Human Rights and Ethics for Clinical Studies (protocol code, E2024-04-001; date of approval, 5 April 2024).

### 2.2. Surgical Procedure

Treatment planning was performed using pre-operative cone beam computed tomography (CBCT) data and wax-up study models. The treatment protocol is illustrated in Figure 1. In three patients with insufficient bone volume, implant placement was performed using a staged approach (St) following the GBR technique (residual bone was less than 1/3 of the length of the implant). Sufficient bone was available in the other three patients for ensuring primary stability following implant placement in the edentulous area. Therefore, these treatment plans were designed to simultaneously perform GBR and implant placement (Si) (residual bone was more than 1/3 of the length of the implant). The other four patients had sufficient bone for ensuring primary stability of the implant placement; however, loss of buccal bone wall was observed following teeth extraction. In these cases, alveolar ridge augmentation beyond the bone housing was necessary; therefore, simultaneous GBR technique with immediate tooth extraction (SiIP) was planned (residual bone after extraction was less than 1/3 of the length of the implant). In all cases, bone-augmentation therapy was performed using CA (Cytrans Granules^®^, M size; GC Corporation, Tokyo, Japan) with P(LA/LC) membranes.

The treatment protocol was as follows: In the case of ST, following mucoperiosteal flap elevation and bone surface debridement, the apical area of the P(LA/LC) membrane was fixed using bone tacks or sutures. CA mixed with saline solution was grafted between the bone surface and half-fixed membrane. After bone grafting, the coronal area of the membrane was fixed, and the grafted CA was tightly packed with an extensible membrane. In addition, in these cases, CA was filled through the membrane gap to the limit of the extensibility of the membrane to avoid micro-movements. This technique was based on that by Urban et al. (2015) [6]. In the case of Si, following mucoperiosteal flap elevation and bone surface debridement, implant placement was performed; the apical area of the P(LA/LC) membrane was fixed using bone tacks or sutures. CA mixed with saline solution was grafted between the bone surface and half-fixed membrane. After bone grafting, the coronal area of the membrane was fixed, and the grafted CA was tightly packed with an extensible membrane. In addition, in these cases, CA was filled through the membrane gap to the limit of the extensibility of the membrane. In the case of SiIP, full thickness flap elevation was performed at the treatment sites following local anesthesia administration. Following tooth extraction, granulation tissue debridement and implant placement were performed; the apical area of the P(LA/LC) membrane was fixed using bone tacks or sutures. CA mixed with saline solution was grafted between the bone surface and half-fixed membrane. After bone grafting, the coronal area of the membrane was fixed, and the grafted CA was tightly packed with an extensible membrane. In every case, after fixed membrane, tension-free wound closure was achieved by conventional invasive periosteal-releasing incision using a scalpel. All sutures were closed with simple sutures using 5–0 nylon sutures. Post-operative medication was administered; cephem as antimicrobial and loxoprofen sodium as anti-inflammatory drug three times a day for four days, and sutures were removed after confirming adequate wound healing at approximately 2–3 weeks postoperatively.

In the case of the staged approach, 6–7 months after GBR, regenerated bone samples were obtained from drilled implant holes for histological evaluation from two patients who provided written consent. Each implant demonstrated sufficient primary stability (>20 N·cm).

### 2.3. Measurements Using CBCT

In every case, dimensional changes in the augmented bone tissue were measured baseline and post-operative CBCT data. For clinical measurement, implant placement simulations were performed using computer simulation software (Landmark system, iCAT, Osaka, Japan), and the pre-surgical bone width (pre-BW) was measured on the pre-surgical CT image. Subsequently, in the case of St, post-surgical CT was performed to fix the implant positions 6 months after GBR. The simulated implant position on the pre-surgical CT image was reproduced on the post-surgical CT image to pinpoint the position of the pre- and post-operative surgical areas, and the post-surgical bone width (Po-BW) was measured. The Po-BW was defined as the longest line perpendicular to the principal axis of the implant in the regenerated bone (Figure 2). In the case of Si and SiIP, two points were decided as the points that were expected to have the least change preoperatively and postoperatively. The point located on the coronal site was designated as rc, and the point located on the apical site was designated as ra. The line connecting rc and ra was designated as the r-line and was used for overlapping the preoperative and postoperative CT images. The CT images were manually overlapped, and the determination and confirmation of the same site was performed by a different measurer. The regenerated bone width (RBW) was calculated at each treatment site (Table 2). In other cases, post-surgical CT was performed immediately before surgery. Pre- and post-operative CT images were overlapped using anatomical characteristics such as residual teeth and bone form. Following Po-BW measurement, including and excluding the implant diameter, the RBW was calculated using overlapped CT images (Figure 2).

### 2.4. Clinical Measurement

Bone quantity achievement (BQA), regenerated bone quality (RBQ), and pre-/postoperative oral vestibular condition (OVC) were evaluated using a five-point scale during the uncovering (Table 3). The BQA was evaluated as the percentage of the predicted preoperative bone regeneration as follows: (1) very poor, less than 20%; (2) poor, approximately 40%; (3) moderate, approximately 60%; (4) good, approximately 80%; and (5) very good, approximately 100%. RBQ was assessed as follows: (1) very poor, no granules remodeled to the bone; (2) poor, granules were partially remodeled to bone but a large amount of granules remained; (3) moderate, granules were largely remodeled to the bone; (4) good, a few granules remained at the surface; and (5) very good, granules were completely remodeled to the bone. OVC was assessed as follows: (1) very narrow, free gingival graft treatment was needed in the uncovering; (2) narrow, although free gingival graft treatment was necessarily not required, the flap required apical positioning; (3) moderate, soft tissue management was not necessary; (4) good, punch-out only; and (5) very good, required soft tissue reduction if necessary. In addition, post-surgical complications involving the peri-implant tissues were assessed. This evaluation method was based on that by Taniguchi et al. [16].

### 2.5. Histological Evaluation

The bone specimens that contained only regenerated bone were collected from the implant placement site using a 2 mm diameter trephine bur (Figure 3g,h). The bone cores were fixed in a paraformaldehyde solution (4% Paraformaldehyde Phosphate Buffer Solution, Fujifilm Wako Pure Chemical Corp., Osala, Japan). The collected samples were immersed in 4% paraformaldehyde overnight and demineralize with formic acid (5% Formic Acid (for Decalcification), Fujifilm Wako Pure Chemical Corp., Osala, Japan). The demineralized samples were embedded in paraffin (PathoprepR 546, Fujifilm Wako Pure Chemical Corp., Osala, Japan), and the paraffin-embedded samples were sliced longitudinally at the center. The histological sections were stained with hematoxylin–eosin and subjected to histometric evaluation under a light microscope (ZEISS Axioscope, Carl Zeiss AG, Oberkochen, Germany). Observations of the tissue sections were performed by two measurers to ensure uniform consistency of each tissue. Tissue sections were observed at 20× and 50× magnification. The area of the new bone was measured on the tissue sections using Image J software (https://imagej.net/ij/ (accessed on 18 December 2024)), and the ratio of the area of the new bone to the area of the entire section was calculated.

### 2.6. Statistical Analysis

The pre- and post-operative bone width data were subjected to a linear mixed-effects model because the test results showed that both pre- and post-operative and regenerated bone widths demonstrated homoscedasticity. The results of all procedures were compared by Welch’s t-test, and for differences in surgery, a t-test was used to compare the difference in the least-squares means between each condition obtained with the linear mixed model (*p* < 0.001).

## 3. Results

No complications, such as infection, membrane exposure, or leakage of bone graft substitutes, were observed post-operatively. Each implant demonstrated osseointegration during the uncovering surgery 4–5 months following implant placement. Following prosthetic treatment, maintenance was observed in all cases.

The mean pre-BW, post-BW, and RBW were 2.81 ± 1.67 mm, 9.14 ± 2.25 mm, and 4.14 ± 1.99 mm, respectively. The mean RBW including the implant diameter of each surgical procedure were Si, 6.34 ± 2.64 mm and SiIP, 7.55 ± 1.17 mm. In the case of Si and SiIP, the mean RBW excluding implant diameter were St, 5.57 ± 1.08 mm; Si, 2.60 ± 2.42 mm; and SiIP, 3.90 ± 0.78 mm (Table 2). Significant differences were observed in the pre- and post-operative bone width, in all the cases and the SiIP group (*p* < 0.001). In all the cases, BQA was achieved for implant placement. The RBQ was evaluated for hardness during hole drilling and primary stabilization of the implants, and the mean RBQ score was 5. The OVC tended to narrow, with a mean score of 3; however, in all the cases, free gingival graft and apically positioned flap surgery were not necessary (Table 3). Prosthetic treatment was performed 1 month following the uncovering surgery.

In case 1, which was performed using St, GBR was performed at sites 11 and 12 and simultaneous ridge preservation was performed at sites 21 and 22 without a membrane (Figure 3). At the ridge preservation site, severe inflammation of the periodontal tissue was attributed to vertical fractures of the tooth, resulting in buccal bone loss. Site 21 had severe inflammation of the periodontal tissue, and, in addition, the mucosal condition was not favorable with residual sinus tracts. Therefore, only CA was grafted to avoid membrane infection due to postoperative wound dehiscence. Six months following CBCT imaging of the GBR site, bone volume was augmented by nearly 5 mm beyond the bone housing in implant sites 12 and 11, and the extraction sockets of 21 and 22 were filled with regenerated bone (Figure 4). Six months after GBR, during implant placement surgery, regenerated bone samples were harvested from the drilled implant holes at sites 12 and 22. Upon histological examination, the tissue of the GBR site demonstrated new bone formation remodeled from the CA and slight residual CA particles. In addition, bone marrow tissue formation was confirmed, indicating functional bone regeneration (Figure 5). On the other hand, new bone was observed in the ridge preservation site. However, the amount of residual CA particles tended to be larger than those in the GBR area. No bone marrow was observed (Figure 6). In addition, sections of bone augmentation revealed that sufficient new bone formation was observed without the use of autogenous bone admixture. Histologic evaluation showed that 52% of the area in case 1 and 42% in case 2 was remodeled into new bone in the tissue sections at the GBR site (Figure 7). Tissue sections in both cases also showed no soft tissue ingrowth in the region of the grafted CA. In case 1, marrow formation was observed in the area of the grafted CA. On the other hand, 13.6% of the area without the membrane in case 1, which was treated with alveolar ridge preservation, showed new bone formation. Although new bone formation could be observed, soft tissue ingrowth was also observed in some areas.

In the case of Si, as indicated in case 8, site 46 showed poor healing after tooth extraction (Figure 8). After 5 months, bone augmentation was observed. The SiIP case is illustrated as Case 9. Significant bone loss was observed at the buccal side (Figure 9). After four months, the peri-implant bone was completely augmented.

## 4. Discussion

The success of implant placement depends on the quality and volume of bone in the implant sites. GBR is necessary to regenerate the bone sufficiently in the implant sites. This case series demonstrated that the CA and P(LA/LC) membranes are safe and effective for GBR procedures in implant treatment. According to Taniguchi et al., in the case of ridge augmentation using CA alone, 3.4 ± 2.3 mm bone width excluding implant diameter was gained in 17 sites [16]. The mean RBW was 3.5 ± 1.9 mm in staged ridge augmentation. Beitlitum et al. performed GBR with a collagen membrane on a mixture of allograft and autologous bone in 50 patients. The average bone augmentation was approximately 5 mm horizontally with allograft and 3.6 mm with autologous bone. Von Arx et al. reported an average horizontal augmentation of 4.6 mm for GBR using autologous block bone graft combined with BBM and a collagen membrane [18,19]. In this case series, 6.36 ± 1.83 mm bone width including implant diameter was gained using CA and the P(LA/LC) membrane in 15 sites. All procedures demonstrated a significant increase in post-operative bone width compared to pre-operative bone width. Only the SiIP group showed a significant difference compared to the pre-operative bone width. These differences were attributed to the narrower pre-operative bone width in the SiIP group than in the other groups; the post-operative bone width was approximately 9 mm in all the sites. Thus, the regenerated bone width is dependent on the pre-operative bone width. Moreover, compared to the other horizontal ridge augmentation technique, the mean RBW was reportedly 4.9 ± 3.7 mm in the Erbium-doped yttrium-aluminum-garnet laser-assisted bone regenerative therapy (Er-LBRT) technique using bovine bone mineral [20]. Er-LBRT has been proposed as an alternative to GBR in recent years; however, the borderline of indications between Er-LBRT and GBR are unknown. In the Er-LBRT case report, although the clinical outcomes are inferior to GBR, the simplicity of the procedure is emphasized. However, simple comparisons between GBR and Er-LBRT are difficult to make as the number of sample sites in the Er-LBRT case report was smaller than that in this case report, and different bone graft substitutes (bovine bone mineral) were used in the Er-LBRT case report. Future research with an even larger sample size using the same materials is warranted to validate the findings of this study. On the other hand, the CT measurements were performed by overlapping the same area; however, there is a possibility that errors may exist in some areas due to the longest length of RBW being measured. Three dimensional volumetric measurement may be necessary in future studies to solve this problem.

According to Taniguchi et al., there were some residual CA particles on the surface of augmented bone that solely used CA. In Figure 3, some residual particles similarly remained in the augmented bone. Furthermore, the high Hounsfield unit value of CA in CT indicates a large amount of CA remaining within the augmented bone. In this study, histological evaluation was performed to ensure the quality of augmented bone and the tendency of residual CA particles. Histological evaluation indicated that the GBR sites contained more new bone than the non-membrane sites. Bone marrow formation was also observed at the GBR site, indicating more functional bone formation. On the other hand, in the bone graft alone site, a slight fibrous tissue ingrowth was observed, and incomplete bone regeneration was apparent. With the histological sections of the present study, it is difficult to discuss the possibility that fibrous tissue will be replaced by bone in the future. These results indicate that the P(LA/LC) membrane has adequate barrier function. In the subjective evaluation, no difference was observed between the GBR site and the bone graft alone site, and the implants also achieved adequate secondary stability; therefore, the difference in bone quality between the use and non-use of the membrane was not a clinically significant issue. However, in each histological section, some residual CA particles were observed. The residual CA particles tended to be remodeled in the deeper areas by more new bone. The results revealed that there was less residual and more new bone formation in the deeper areas than the residual amount of particles on the augmented bone surface. On the other hand, in this study, bone augmentation was performed to place the implants to the optimal position; therefore, the implant may not be supported only by the augmented bone. Residual CA particles may be remodeled to bone tissue as time passes. In Figure 3k, uncovering surgery was performed 9 months after GBR. During the uncovering surgery, no residual CA particles were observed in the augmented bone surface. Moreover, in Figure 8d and Figure 9d, CA particles were not observed near the residual bone. Therefore, CA tends to remodel to bone more rapidly in areas closer to the residual bone. In a previous GBR bone augmentation histological evaluation report, Ersanli et al., in 2004, showed that GBR with BBM and collagen membrane in 11 patients showed histological results of 46% new bone formation in the maxilla and 63% in the mandible after an average follow-up period of 7 months [21]. According to a 2016 report from Bassi et al. with GBR using titanium foil and allogeneic bone paste, bone samples were collected after approximately 6 months for histological evaluation. The results showed approximately 48% new bone formation [22]. In addition, Čandrlić et al. showed that GBR using a mixed material of hydroxyapatite and β-TCP in combination with a collagen membrane resulted in histologically observed new bone formation of approximately 25% [23]. Although it is difficult to make a simple comparison, the clinical results of GBR with a P(LA/LC) membrane and CA are considered equivalent to those of conventional materials when compared to the rate of new bone formation in this study. However, in this study, clinical data were collected using three different treatment procedures in a limited number of 15 cases; therefore, studies with a larger sample size are warranted to improve the accuracy of the data.

The BQA and RBQ demonstrated good outcomes in all the procedures. However, the OVC declined in the pre-operative and post-operative periods. This is because GBR requires a large releasing incision for wound closure; in particular, SiIP, which is performed simultaneously with tooth extraction, is thought to significantly reduce the size of the oral vestibule during suturing of the extraction socket.

## 5. Conclusions

The GBR technique using CA alone and P(LA/LC) membranes achieved favorable clinical outcomes for the Si, SiIP, and St procedures. Histological evaluation in St cases revealed that new bone formation was observed in GBR with the P(LA/LC) membrane. These results suggested that GBR with membrane and CA was clinically sufficient for bone augmentation in implant therapy.

## Figures and Tables

**Figure 1 dentistry-13-00085-f001:**
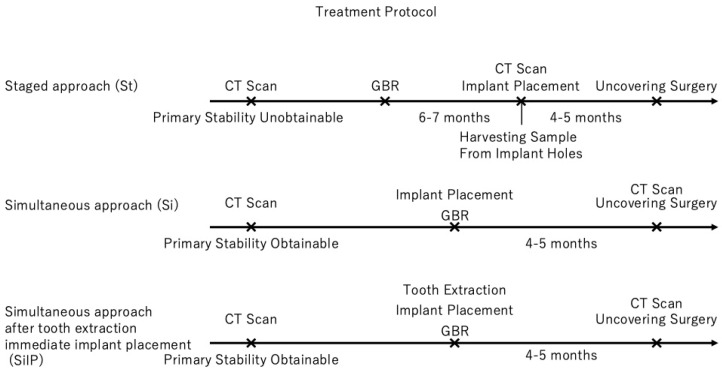
Treatment procedure of each surgical procedure.

**Figure 2 dentistry-13-00085-f002:**
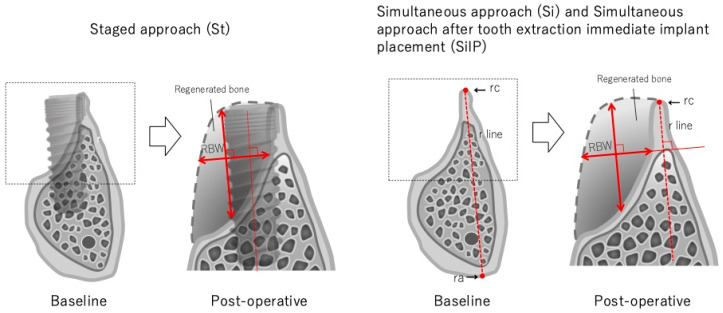
CBCT measurement procedure of each procedure. RBW: regenerated bone width, rc: reference point of coronal area, ra: reference point of apical area, r line: reference line.

**Figure 3 dentistry-13-00085-f003:**
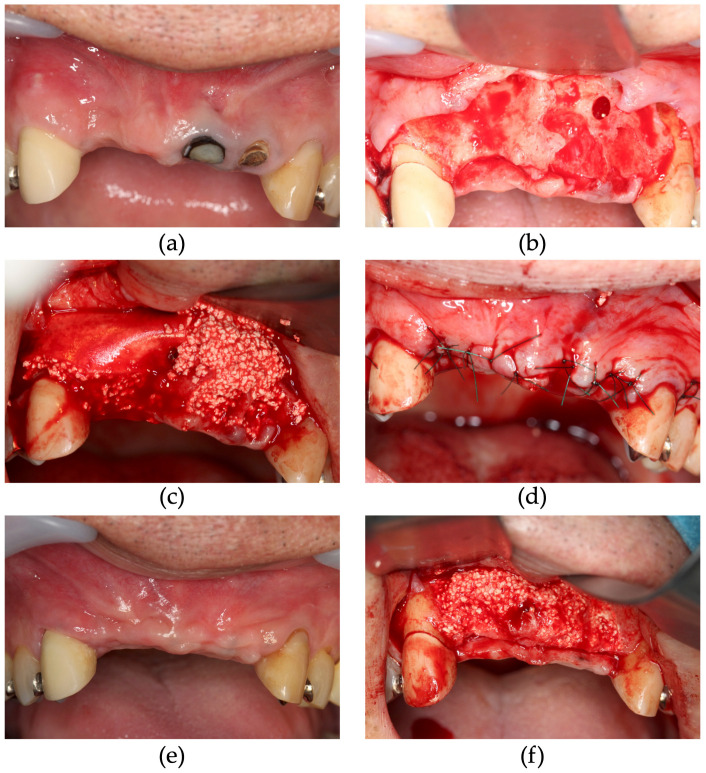
Staged approach of implant placement with carbonated apatite (CA) and poly (L-lactic acid/ε-caprolactone) (P(LA/LC)) membrane. (**a**) Pre-operative view, the bone amount of the 11 and 12 area was not sufficient. (**b**) After flap reflection, the teeth and granulation tissue were removed. (**c**) During guided bone regeneration, CA and the P(LA/LC) membrane were placed at the 11 and 12 sites. CA was placed at the 21 and 22 sites without membrane. (**d**) Immediately after operation, the wound was closed using a tension free suture. (**e**) No complications were observed 6 months after GBR immediately before implant placement. (**f**) Six months after GBR, at the implant placement surgery, the alveolar ridge was augmented. (**g**) The GBR bone specimens were collected from 12 implant hole. (**h**) The CA alone bone specimens were collected from 21 implant hole. (**i**) After implant placement, implants were placed in the bone augmentation area. The arrow indicated the regenerated bone harvested site. (**j**) Four months after implant placement, before uncovering surgery. (**k**) Implants were covered by regenerated bone. There were no residual CA particles. (**l**) Three weeks after uncovering the surgerical area, the super structure was fastened.

**Figure 4 dentistry-13-00085-f004:**
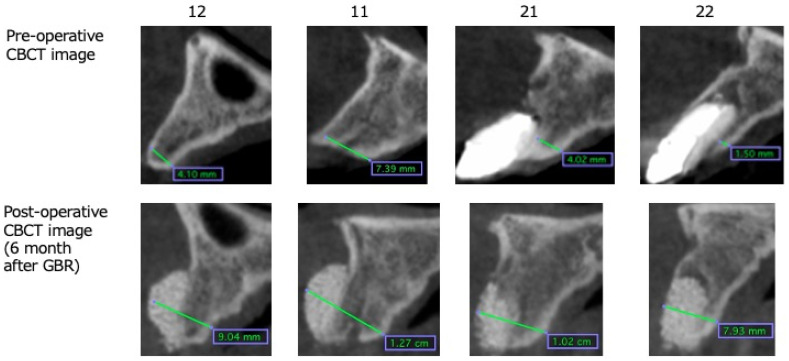
Comparison of the pre- and post-operative cone-beam computed tomography crosscut image. In the 12 and 11 sites, bone volume was augmented nearly 5 mm beyond the bone housing, and in the 21 and 22 sites, extraction sockets were filled regenerated bone.

**Figure 5 dentistry-13-00085-f005:**
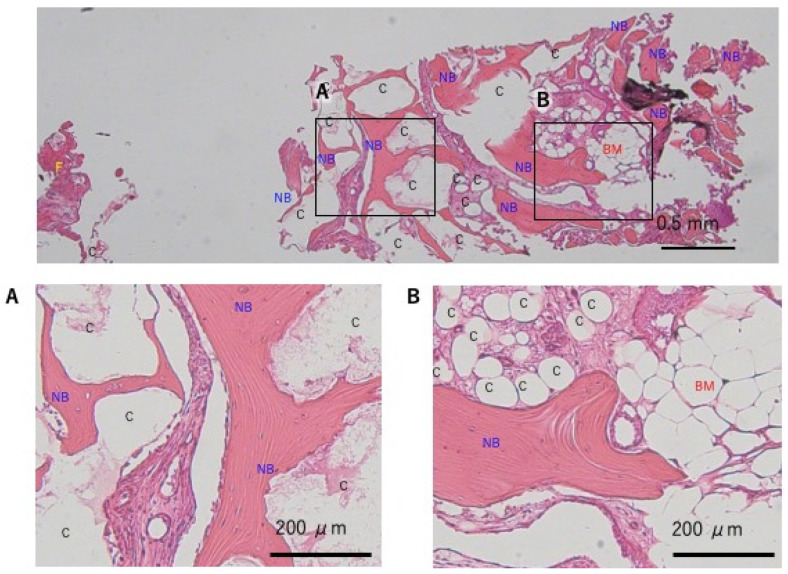
Case 1 12 site, histological evaluation at the guided bone regeneration site: (**A**) granules of bone graft material remained; however, new bone formation was observed in most areas. (**B**) bone marrow formation was also observed in some areas. C: carbonate apatite, NB: new bone, BM: bone marrow, F: fiber.

**Figure 6 dentistry-13-00085-f006:**
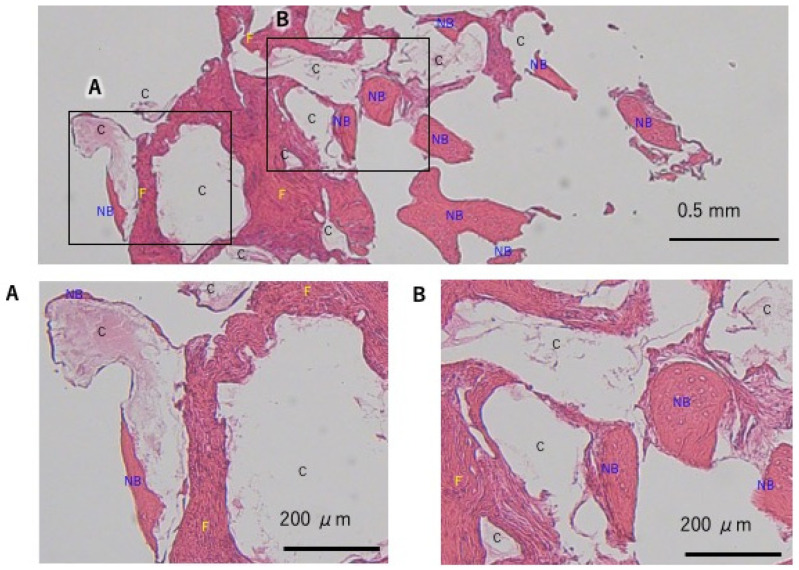
Case 1 21 site, histological evaluation at the ridge preservation site. (**A**) Compared to the guided bone regeneration (GBR) site, fibrous tissue ingrowth was observed, and there was a trend for more residual granules and less new bone formation. (**B**) New bone formation was observed in the deep area. C: carbonated apatite, NB: new bone, F: fiber.

**Figure 7 dentistry-13-00085-f007:**
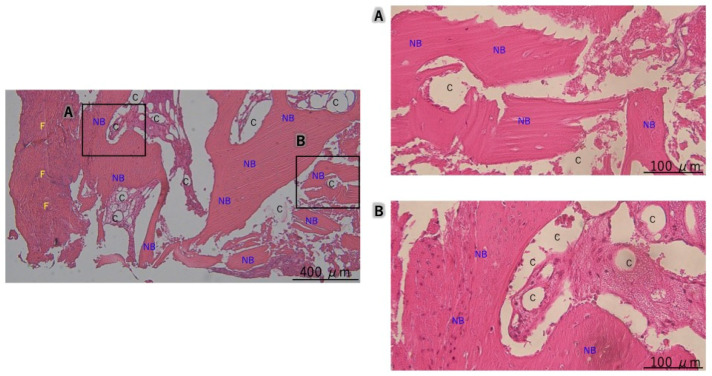
Case 2, Histological evaluation at the guided bone regeneration site: (**A**) New bone formation is observed without soft tissue ingrowth. (**B**) In deeper areas, there is partial residual bone graft material; however, new bone is visible in the majority of the areas. C: carbonate apatite, NB: new bone, F: fiber.

**Figure 8 dentistry-13-00085-f008:**
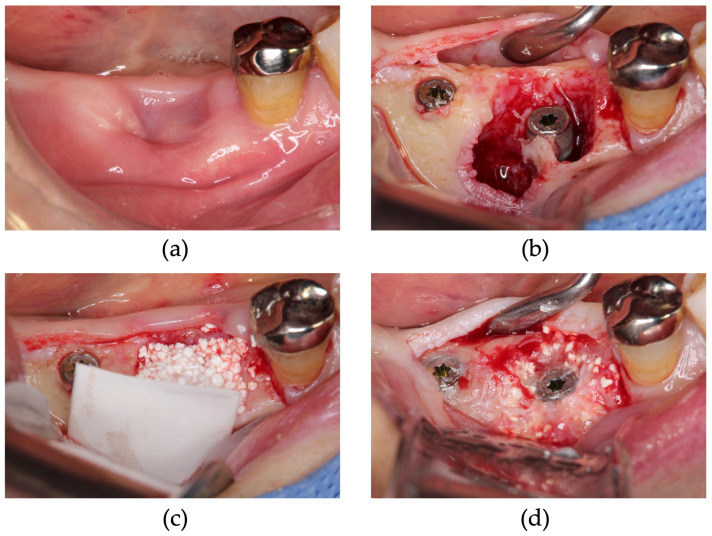
Simultaneous approach of implant placement using carbonated apatite (CA) and poly (L-lactic acid/ε-caprolactone) (P(LA/LC)) membrane. (**a**) Three months after tooth extraction, bone healing was not sufficient. (**b**) After granulation tissue removal, implants were placed. (**c**) CA and the P(LA/LC) membrane were placed at the 46 site. (**d**) During uncovering surgery, the alveolar ridge was augmented.

**Figure 9 dentistry-13-00085-f009:**
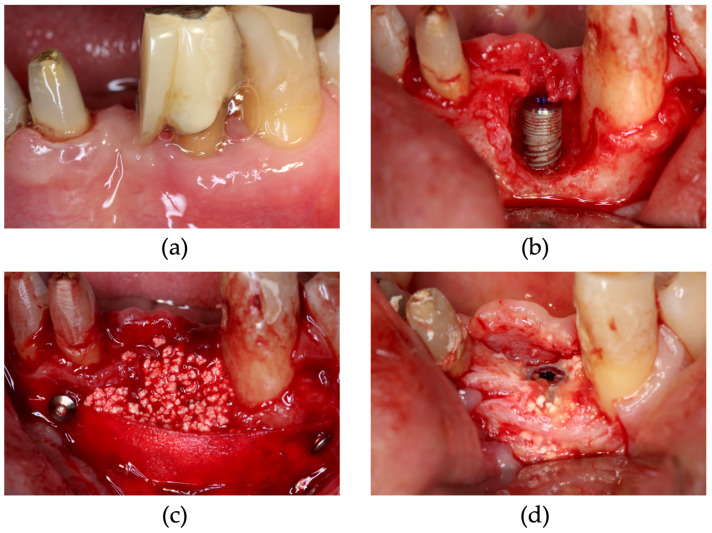
Simultaneous approach and immediate implant placement using carbonated apatite (CA) and poly (L-lactic acid/ε-caprolactone) (P(LA/LC)) membrane. (**a**) Pre-operative view, 32 had a vertical root fracture. (**b**) After tooth and granulation tissue removal, implants were placed. (**c**) CA and the P(LA/LC) membrane were placed at the 32 site. (**d**) During uncovering surgery, the alveolar ridge was augmented.

**Table 1 dentistry-13-00085-t001:** Patient demographics and treatment sites.

Patient Number	1		2		3		4	5	6	7		8	9	10	
Case number	1	2	3	4	5	6	7	8	9	10	11	12	13	14	15
Age (Y O)	63	63	21	21	65	65	54	41	56	54	54	73	61	68	68
Sex (Male = m, Female = f)	m	m	f	F	f	f	M	m	M	f	f	f	m	m	m
Number of treatment sites (FDI)	12	13	12	13	11	12	35	46	37	21	22	46	33	35	36
Procedure	St	St	St	St	St	St	SiIP	SiIP	SiIP	Si	Si	Si	SiIP	Si	Si
implant diameter (mm)	3.3	4	3.3	3.3	3.3	4	3.3	4	4	3.3	3.3	4	3.3	4	4

Age: 55.6 ± 14.3 (Mean ± SD).

**Table 2 dentistry-13-00085-t002:** Comparison of the pre- and post-operative bone width.

	Pre-Operative Bone Width	Post-Operative Bone Width	Regenerated Bone Width Without Implant Diameter	Regenerated Bone Width Within Implant Diameter
	(Mean ± SD mm)	(Mean ± SD mm)	(Mean ± SD mm)	(Mean ± SD mm)
All procedure (N = 15)	2.81 ± 1.67	9.14 ± 2.25 *	4.14 ± 1.99	6.36 ± 1.83
Si (N = 5)	3.06 ± 0.66	9.33 ± 2.48	2.60 ± 2.42	6.34 ± 2.64
SiIP (N = 4)	1.81 ± 1.09	9.37 ± 2.08 *	3.90 ± 0.78	7.55 ± 1.17
St (N = 6)	3.27 ± 2.37	8.84 ± 2.54	5.57 ± 1.08	

SD, standard deviation, Welch’s *t*-test; *p* < 0.001, *, Statistically significant difference in pre- and post-operative data.

**Table 3 dentistry-13-00085-t003:** Bone quality achievement, regenerated bone quantity, and oral vestibule condition evaluation.

	Bone Quantity Achievement (Median)	Regenerated Bone Quality (Median)	Pre-Operative Oral Vestibular Condition (Median)	Post-Operative Oral Vestibular Condition (Median)
All Procedure (N = 15)	5	5	4	3
Procedure				
Si (N = 5)	5	5	4	3
SilM (N = 4)	5	5	4	2.5
St (N = 6)	5	5	3	3

## Data Availability

The paper is self-containing. For additional information or data, please contact the corresponding author.

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
