# Peer review of "Guided Bone Regeneration Using Carbonated Apatite Granules and L-Lactic Acid/ε-Caprolactone Membranes: A Case Series and Histological Evaluation"

_dentistry, 2025, doi:10.3390/dj13020085_

Round 1
Reviewer 1 Report (Previous Reviewer 2)
Comments and Suggestions for Authors
comments in file

Author Response
Reviewer 1
This is a very interesting manuscript about “Guided Bone Regeneration Using Carbonated Apatite Granules and L-lactic Acid/ε-caprolactone Membrane: A Case Series and Histological Analysis”. I consider that this manuscript is interesting, with enough information and well organized for publication in this journal, the next comments are for improve the quality of manuscript.
Thank you very much for your kind peer review. The detailed peer review and reviewer's comments and suggestions have greatly improved the quality of our manuscript. The answers to the points of your suggestions are summarized below.
Introduction section, the information shown is adequate.
Case series, there are a large range between the age in patients (21, 73 years) please justify this used range.
There were a limited number of patients who required GBR and whose consent was obtained during the study period. Therefore, to collect as much data as possible, we analyzed the data in this article within a wide age range.
Table 1, implant diameter, add measure unit (for example, mm, inches) since only appears the numbers.
Thank you for your valuable suggestion. Table 1 has been corrected. (indicated by blue highlights)
Conclusion section, please add a brief text about the clinical importance of this study.
We have added the following sentence into the conclusion to clarify the clinical significance of this manuscript
Lines 366–367: These results suggested that GBR with membrane and CA was clinically sufficient for bone augmentation in implant therapy.
Reviewer 2 Report (Previous Reviewer 3)
Comments and Suggestions for Authors
1. What is the novelty of the work?
2. Line 226: Make a space between images.
3. Line 243: GBR is well-known and useful in implant placement and bone regeneration. What is new about your GBR membrane?
4. Did you compare other commercial membranes? What is special?
5. Poly L-lactic acid /ε-caprolactone are very slow degradable polymers. Why did you mix and choose those polymers?
6. For the histological analysis case, is only HE staining enough? In your title, you mentioned histological analysis, which means it should be a deeper analysis.
Comments on the Quality of English LanguageEnglish is fine. But the explanation was not clear.
Author Response
Reviewer 2
Thank you very much for your kind peer review. The detailed peer review and reviewer's comments and suggestions have greatly improved the quality of our manuscript. The answers to the points of your suggestions are summarized below.
- What is the novelty of the work?
This manuscript provides a clinical evaluation of a newly released bone graft material, CA, and a P(LA/LC) membrane. The amount of evidence for the P(LA/LC) membranes used for bone augmentation is insufficient owing to the paucity of clinical publications on these materials. The novelties in this manuscript are that the case series clearly shows the average augmented bone width of GBR with CA, and a P(LA/LC) membrane, the histological comparison between GBR with and without the P(LA/LC) membrane, and the applicability of these materials to a variety of techniques such as simultaneous approach and staged approach. In addition, there are only a few articles that contain human tissue sections of GBR using CA alone as a bone graft material.
- Line 226: Make a space between images.
We have added space to near the original text (line 226).
- Line 243: GBR is well-known and useful in implant placement and bone regeneration. What is new about your GBR membrane?
- Did you compare other commercial membranes? What is special?
- Poly L-lactic acid /ε-caprolactone are very slow degradable polymers. Why did you mix and choose those polymers?
Questions 3, 4, and 5 are relevant and are answered together below.
This membrane has the characteristics of delayed resorbability and extensibility. Abe et al. reported that the resorption period of this membrane was adjusted to the optimal period for bone augmentation by using P(LA/LC) (reference 10).
Although this study did not compare GBR with other membranes, Shido et al. performed GBR with a bone grafting material mixed with autogenous bone and bone grafting material using collagen and P(LA/LC) membranes and concluded that the effect was comparable to that of collagen membranes (reference 11). In this study, CA and autogenous bone were not mixed for the purpose of minimally invasive GBR. In its new functionality, we consider that it is a fully chemically synthetic material that has the same effect as existing collagen membranes, which resulted in significant GBR with resorbable membranes in patients who detest the use of bovine and other biologically derived materials.
The following sentence has been added to clarify the effect of P(LA/LC).
Lines 61–62: The resorption period has been reported to be optimal for bone augmentation by using P(LA/LC) as the material.
- For the histological analysis case, is only HE staining enough? In your title, you mentioned histological analysis, which means it should be a deeper analysis.
After consideration, we have decided to change the title of the paper from Histological Analysis to Histological Evaluation. The title and words in the article have been corrected accordingly. We have indicated the revised text in blue highlights.
Reviewer 3 Report (New Reviewer)
Comments and Suggestions for Authors
In this manuscript, the discussion is clear and the structure of the article is clear. This article is intended to start with clinical problems and has certain innovations. Overall, we have evaluated in detail the effect of implant treatment for alveolar ridge augmentation using bone grafts composed of P (LA/LC) membrane and carbonate apatite and confirmed that implant treatment using P (LA/LC) membrane and carbonate apatite as GBR materials leads to stable and good bone regeneration. However, combined with the experimental purpose and experimental results, this article still needs to be significantly revised. Supplementing and improving the following content is recommended before the publication of this article.
1. It is recommended that the capitalization of "f/m" in the "" column in Table 1 be consistent;
2. The assessment of alveolar bone volume for each patient in the "Case Series" is recommended to be supplemented, or the specific reasons for the grouping are explained. Since the expression of this part is different from the preoperative bone volume in Table 2, it is recommended that it be improved.
3. Why is Row st in Table 2 incomplete?
4. Whether the age of the included cases in the st group is too old to affect the judgment of the bone regeneration results;
5. In Figure 3, it is recommended to add text to intuitively explain the specific surgical process;
6. It is recommended to add a scale in Figure 4;
7. The experimental results show that SiIP with good initial stability has a good postoperative treatment effect, but the bone condition of this experimental group is good, and whether the experimental results can fully explain its role in promoting bone formation is explainable.
Author Response
Reviewer 3
In this manuscript, the discussion is clear and the structure of the article is clear. This article is intended to start with clinical problems and has certain innovations. Overall, we have evaluated in detail the effect of implant treatment for alveolar ridge augmentation using bone grafts composed of P (LA/LC) membrane and carbonate apatite and confirmed that implant treatment using P (LA/LC) membrane and carbonate apatite as GBR materials leads to stable and good bone regeneration. However, combined with the experimental purpose and experimental results, this article still needs to be significantly revised. Supplementing and improving the following content is recommended before the publication of this article.
Thank you very much for your kind peer review. The detailed peer review and reviewer's comments and suggestions have greatly improved the quality of our manuscript. The answers to the points of your suggestions are summarized below.
It is recommended that the capitalization of "f/m" in the "" column in Table 1 be consistent;
We have unified the denotation of f and m. (indicated in blue highlights)
The assessment of alveolar bone volume for each patient in the "Case Series" is recommended to be supplemented, or the specific reasons for the grouping are explained. Since the expression of this part is different from the preoperative bone volume in Table 2, it is recommended that it be improved.
Because the preoperative bone width differs from the bone width required for ideal implant placement, the data show less residual bone width for St than for SiIP. The following text has been added to clarify these points.
Lines 101–102: (Residual bone was less than 1/3 of the length of the implant)
Line 105: (Residual bone was more than 1/3 of the length of the implant)
Lines 109–110: (Residual bone after extraction was more than 1/3 of the length of the implant)
Why is Row st in Table 2 incomplete?
Because the measurement period was before implant placement, St group was not required to state the diameter of the implant in table 2.
Whether the age of the included cases in the st group is too old to affect the judgment of the bone regeneration results;
Based on the histologic evaluation, age was not considered a significant factor in this study because the percentage of new bone formation was higher in Case 1. However, we were unable to arrive at a definitive conclusion owing to the small number of cases.
In Figure 3, it is recommended to add text to intuitively explain the specific surgical process;
The following text has been added for clarity.
Line 260: During guided bone regeneration,
Line 261: Immediate after operation.
Line 264: After implant placement.
It is recommended to add a scale in Figure 4;
Figure 4 shows the actual CT measurements; therefore, we did not consider it necessary to add a scale.
The experimental results show that SiIP with good initial stability has a good postoperative treatment effect, but the bone condition of this experimental group is good, and whether the experimental results can fully explain its role in promoting bone formation is explainable.
We suggest that SiIP may have resulted in more amount of bone augmentation because of the narrower baseline bone width than other techniques. This is different from biological reasons. SiIP results in the largest increase in postoperative bone width; however, the average difference is less than 0.1 mm when compared to Si. In addition, St has the highest regenerated bone width without implant diameter; therefore, it is difficult to conclude that SiIP provides the best result. Owing to these reasons, it is difficult to conclude that only SiIP is superior. Thus, all St, Si, and SiIP procedures can be considered effective.
This manuscript is a resubmission of an earlier submission. The following is a list of the peer review reports and author responses from that submission.
Round 1
Reviewer 1 Report
Comments and Suggestions for Authors
Comments:
This manuscript presents cases treated with carbonated apatite with/without P(LA/LC) membrane. The effort to present the radiographic and histological outcomes related to these materials is commendable. However, there are several clarifications needed before publication of this manuscript. Among others, the rationale for studying these materials, surgical and measurement methodology is unclear. Specific comments can be found below.
Title
· The tile mentions “case study” when 10 patients with 15 sites were studies, I recommend changing to case series. Also, change phrase case study and case report to case series throughout the manuscript.
Abstract
Methods:
· Please only mention number of sites per surgical technique.
· How long was the follow up?
· Histological analysis is mentioned in the title but not in the abstract, please add some histological results in the abstract.
Introduction
· The first 2 sentences on periodontal regeneration are not relevant to the paper.
· I disagree with the statement that resorbable membranes provide unpredictable GBR outcomes. There are several systematic reviews and meta-analyses with robust GBR outcomes. Please discuss those.
· Please change citation Urvan et al. to Urban et al. The sausage technique is the only one described in the introduction, but not mentioned anywhere else in the manuscript. From the methods description, I assume that this is the surgical technique used for the cases. If this is correct, please add it in the methods section.
· Additional details on the materials of the studies and citing preclinical and clinical studies is necessary. Why were these materials studied? Why should we learn more about them?
Materials and methods
Case study:
· Please move patient number, description and surgery group (Table 1) to results.
Surgical procedure:
· The only treatment protocol that is described is the one for immediate implant placement. Please describe surgical protocols for all groups.
· “After bone grafting, the coronal area of the membrane was fixed, and the grafted CA was tightly packed with the an elastic membrane”. It is unclear what technique was used for membrane fixation on the buccal. The term “packed with an elastic membrane” is confusing. Was there another membrane?
· Please specify antimicrobial and anti-inflammatory drugs prescribed postoperatively, as well as suture design and material.
· The timeline of treatment is well described in Figure 1. I recommend enriching the previous paragraphs with surgical techniques (implant stability, technique of bone core harvesting) and the graph with the timing of bone core harvesting and removing the last paragraph of this section.
Clinical measurements:
· Bone width was measured in CBCTs, therefore the title of the section is not relevant. Please change accordingly.
· There is insufficient information on bone measurement technique. Who did the measurements? Was/Were the examiner(s) calibrated? Were the CBCTs superimposed (overlapped images were mentioned without more information)? Where the measurements done manually or automatically by a software? At what level from the crest was bone width measured? Was bone height measured? How?
· A modified detailed figure 3 with the radiographic analysis would be useful.
· “Bone quantity achievement (BQA), regenerated bone quality (RBQ), and pre-/post- 135 operative oral vestibular condition (OVC) were evaluated using a five-point scale during 136 the second-stage surgery according to Taniguchi et al [11].” Please provide definitions and describe this evaluation method further as these term are not usually used. If these are the clinical measurements you are referring to, please specify and keep under “clinical measurements” section.
Histological analysis:
· There are minimal details on histological analysis. Who did the analysis and reading of the slides? Calibration considerations?
Statistical analysis
· The sample size is small. Are the data normally distributed? Should non-parametric tests be used?
· Are the data presented as mean±SD? SE? In table 3, a median was mentioned.
· The only statistical test mentioned is the linear mixed effects model, however in results there are p-vlaues presented for between-group comparisons. What tests were used for this comparison?
· RBH and RBW was abbreviation mentioned for the first time in this section. Assuming that these are Ridge Bone Height and Rigde Bone Width, there is minimal description of the width measurement methodology and no mention of the height measurement methodology. Also, RP and RA are abbreviations first mentioned here. RP and RA are used In a linear mixed effects model, apart form now knowing what RA an RP (therefore not being able to evaluate the model), how can a study with 10 patients and 15 sites have
Results
· The results present pooled data from all patients together and per group in the first paragraph. The second paragraph describes one of the cases with the histological data. I agree that the description of defects and photographs are important for the subjects contributing to the histological analysis, however, this paragraph is repetitive with Figure 1 and the surgical procedure paragraph of materials and methods. I recommend that only the defect description remains.
· In previous sections, it is mentioned that two patients contributed with sites for the histological analysis. How many sites did they contribute to? In the results, only the histological data (case 1) from one patient (2 sites) are included. Additionally, the histological data are poorly presented, without reporting % for each type of tissue. In the discussion and figures, we can see that there are histological data from sites with and without membrane, in my opinion, these results cannot be pooled or compared and add much inconsistency.
· The achievement of osseointegration is repeated in results.
Figure 2:
· “CA and P(LA/LC) membrane were placed at the 11 and 12 sites. CA was placed at the 21 and 22 199 sites without membrane.” Adding bone graft on the buccal of a ridge without a membrane is not GBR. Why was a membrane used on the right side but not the left side? The fact that a bone core was harvested from site #22 and histological results are presented adds bias to the results, as this is not GBR as mentioned in the title.
Figures 4 and 5:
· Please add tooth numbers for each site so that the reader can correlate with clinical presentation.
Discussion
· The results of this study are not comprehensively compared (clinically, radiographically and histologically) with others that are considered the gold standard.
· Instead, the study results are compared with the Erbium-doped yttrium-aluminum-garnet laser- assisted bone regenerative therapy (Er-LBRT) technique using bovine bone mineral. This is an uncommon technique and it is the first time in the manuscript that this technique is mentioned. I would recommend that the results of the study are compared with more “traditional” techniques, as the use of materials in this case series is not the gold standard, neither is the comparison with Er-LBRT one of the goals of this case series.
Conclusions
· In my opinion, the conclusions are misleading. There are multiple areas in the manuscript where it is mentioned that not all cases received a membrane with unclear distribution between sites. A more detailed, yet concise, description of sites and results is more relevant.
Comments on the Quality of English LanguageThe manuscript should be reviewed by a native English speaker. Limited examples for modification are below: Enormous bone defects, the peri-implant bone was completely augmented, the superstructrure was fastened
Reviewer 2 Report
Comments and Suggestions for Authors
comments in word

Reviewer 3 Report
Comments and Suggestions for Authors
1. What is the novelty of this work?
2. What is different compared to other membranes?
3. GBR materials are helpful for bone growth,h which is already well known. So, I cannot see any difference.
4. A case study, but what is the purpose of this work?